# An Experimental and Virtual Approach to Hip Revision Prostheses

**DOI:** 10.3390/diagnostics12081952

**Published:** 2022-08-12

**Authors:** Alina Duta, Dragos-Laurentiu Popa, Daniela Doina Vintila, Gabriel Buciu, Nicolae Adrian Dina, Adriana Ionescu, Mihaela Corina Berceanu, Daniel Cosmin Calin

**Affiliations:** 1Faculty of Mechanics, University of Craiova, 200512 Craiova, Romania; 2Faculty of Nursing, Titu Maiorescu University, 210102 Targu Jiu, Romania; 3State University of Physical Education and Sport, MD 2024 Chisinau, Moldova; 4University of Medicine and Pharmacy of Craiova, 200349 Craiova, Romania; 5County Emergency Hospital of Slatina, 230008 Slatina, Romania

**Keywords:** hip joint, prostheses, finite element method, human hip revision, virtual test

## Abstract

(1) Introduction: The changes in the joint morphology inevitably lead to prosthesis, but the hip pathology is complex. The hip arthroplasty is a therapeutic solution and can be caused, most frequently, by primary and secondary coxarthrosis due to or followed by traumatic conditions. The main aim of this study was to find the method of revision hip prosthesis that preserves as much bone material as possible and has sufficiently good mechanical strength. (2) Materials and Methods: In this study, in a first step, the two revision prostheses were performed on bone components taken from an animal (cow), and then, they were tested on a mechanical testing machine until the prostheses physically failed, and the force causing their failure was determined. (3) Results: These prostheses were then modelled in a virtual environment and tested using the finite element method (FEM) in order to determine their behaviour under loading from normal human gait. Displacement, strain, and stress maps were obtained. (4) Discussion: Discussions on hip revision prostheses, method, and theory analysis are presented at the end of the paper. (5) Conclusions: Important conclusions are drawn based on comparative analyses. The main conclusion shows that the both orthopaedic prostheses provide a very good resistance.

## 1. Introduction

From a biomechanical point of view, the human hip joint is spherical; therefore, it can allow rotations on all spatial axes of movement and has an essential importance in the locomotion and statics of the human body.

The changes in joint morphology inevitably lead to prosthesis, but the hip pathology is complex. The hip arthroplasty is a therapeutic solution and can be caused, most frequently, by primary and secondary coxarthrosis due to or followed by traumatic conditions. The femoral fracture and pseudarthrosis of the femoral neck as the final stage of aseptic femoral head necrosis but also rheumatoid arthritis or ankylosing spondylitis are part of the pathology leading to hip joint replacement [1,2,3].

Endoprosthetic arthroplasty is a reconstructive surgical procedure that involves the prosthetic replacement of joint components and bone loss [4].

There has been an evolution in the treatment of hip joint pathology over the last two hundred years from primitive surgery to modern joint arthroplasty, which has undergone a major development over the last thirty years. Although this surgery is a safe and now classic intervention, it remains a permanent challenge due to the desire to invent the “ultimate prosthesis” that perfectly replaces the human joint. Moreover, due to the increasing number of total hip arthroplasty (THA) surgeries combined with increasing life expectancy, the number of hip replacement revisions is obviously expected to improve [5].

Special features occur during the revision of the hip prosthesis. The hip revision surgery involves a special pathology characterized by bone loss in the acetabular areas, cracks in the pelvic bone, enlargement of the contact area of the prosthesis cup, and often bone loss. The restoration of the hip joint by surgery involves additional challenges in these very special situations. In addition, the revision of hip prosthesis is performed on elderly patients, and special methods and techniques are used to replace, reconstruct, or restore additional metal elements [6,7,8].

With reference to the initial surgical moment, prosthesis loosening consists of a deterioration of the position or functions of a complete hip prosthesis. There are also the notions of “decementation” for cemented prostheses or “loss” [6,7,8,9,10,11]. The loosening is a natural and normal step in the evolution of an arthroplasty and can be influenced by a multitude of factors [12,13]. In order to explain this complex phenomenon, over time, researchers have formulated a number of mechanisms and theories. The mechanisms that cause hip prosthesis loosening have been observed to sometimes overlap or amplify, destroying the contact between the cement and the prosthesis or the bone–prosthesis/bone–cement interface [14,15,16]. Several methods of revision prosthesis and obviously several surgical techniques are known. The main aim of this study was to find the method of revision hip prosthesis that preserves as much bone material as possible and has sufficiently good mechanical strength [17,18,19,20,21].

The following observations were drawn after analysing the models, methods, and techniques obtained in several interdisciplinary studies:-The models can be used in various real or virtual tests and experiments;-The virtual models of the revision components can be attached to the bone components, and various “in vitro” tests can be obtained in the virtual environment;-The prosthetic components are generally parameterized in virtual environments, so they can be adapted to different anthropological dimensions.

In this study, as a first step, the two revision prostheses were performed on bone components taken from an animal (cow), and then, they were tested on a mechanical testing machine until the prostheses physically failed, and the force that caused them to fail was determined. Then, these prostheses were modelled in a virtual environment and tested using the finite element method (FEM) to determine their behaviour under loading determined by normal human gait [22]. The prostheses were made by an orthopaedic surgeon.

The present study was approved by the Ethics Committee of the University of Medicine and Pharmacy of Craiova, Romania (approval reference no. 81/23 September 2020), in accordance with the ethical guidelines for research with human and animal participants of the University of Medicine and Pharmacy of Craiova, Romania. Real and virtual experiments were developed in some Laboratories of the Faculty of Mechanics, University of Craiova.

In this study, a virtual method was validated through a real experiment for the analysis of different orthopaedic revision systems of the hip joint, and a technique was highlighted by which bone sacrifice is reduced.

## 2. Material and Method

### 2.1. Material and Method for Performing Orthopedic Hip Revisions

Four hip joints taken from an animal (cow) were used to obtain the revision prostheses. Of these, only two were used to obtain the orthopaedical revision prosthesis with titanium augment and the prosthesis with morcellated graft and reconstructive mesh. These were procured from a specific food industry abattoir. In addition, the prostheses were made no later than six hours after their preservation so as to preserve their mechanical and physical properties. Cow hip joints were used because the acetabular and femoral head dimensions are similar to human ones (Figure 1). Their mechanical properties are also considered to be similar to human ones [23,24].

Specific instruments for orthopaedic surgery were used in order to obtain orthopaedic prostheses. Some of these are used for milling the acetabular cavity, fitting to prosthesis components, and simulating bone defects that are associated with specific hip revision pathology. Orthopaedic cement was also used for cementing the prosthetic cups. The orthopaedic cement has two components: a base and a reinforcement component. It is placed on the components that need to be fixed, and the final fixation is achieved after 30 min. The orthopaedic cement was used in the two prostheses that were studied. Standard surgical gloves were used in all operations to make the prostheses.

### 2.2. Material and Method for Revision Orthopaedical Prosthesis with Titanium Augment

The prosthetic elements as well as the additional revision component (titanium augment) that will form the prosthesis, together with the bone component, are shown in Figure 2. The titanium augment will be fixed to the bone component with two orthopaedical screws.

A first step was to adapt the acetabular cavity to the size of the polyethylene cup of the main prosthesis. This operation was performed with a special set of orthopaedical drills, and the steps of this operation are shown in Figure 3. Initially, the acetabular cavity was widened, and then, portions of bone were removed so that the polyethylene cup of the prosthesis would fit as correctly as possible in the cavity of the pelvic bone. The bone loss encountered in most cases during hip revision surgery was also simulated.

An additional hemispherical cavity was milled into the bone component for the use of the titanium augment. Some steps of this operation are shown in Figure 4. At the end of this stage, the polyethylene cup and additional revision element were positioned on the iliac bone.

To obtain the orthopaedic prosthesis, an orthopaedic cement was used, which has two components, namely a base and a reinforcement. The two components were mixed, and then, the titanium augment was fixed, followed by the polyethylene cup of the prosthesis. Finally, the prosthesis components were pressed into the cavities created in the iliac bone. The specific instruments for these operations were also used. These steps are shown in Figure 5.

This prosthesis is to be placed on a universal test machine. After the prosthesis is made, because the working height of the machine is limited to 27 cm, the orthopaedic prosthesis had to be adapted, i.e., cut, and these operations are shown in Figure 6.

### 2.3. Material and Method Fororthopaedical Prosthesis with Morcellated Bone Graft and Reconstructive Mesh

In a first phase, a morcellated bone graft was obtained from the femoral head area using orthopaedic instruments, as shown in Figure 7.

Areas to be removed were marked on the iliac bone taken from the animal to simulate the bone loss encountered in hip joint revision pathology, as shown in Figure 8.

These marked areas were removed using orthopaedic instruments, and the positioning of the polyethylene cup was tested, as shown in Figure 9.

A morcellated bone graft was placed in the acetabular cavity, and it was pressed into areas of bone loss using specific orthopaedic instruments, as shown in Figure 10.

Next, the reconstructive mesh was prepared and trimmed and adjusted to the required dimensions to cover the morcellated bone graft, as shown in Figure 11.

In the next step, the reconstructive mesh was fixed using orthopaedical screws, and the positioning of the polyethylene cup was checked, as shown in Figure 12.

Finally, the acetabular cavity was covered with orthopaedic cement, and the morcellated graft and reconstructive mesh prosthesis was obtained, as shown in Figure 13.

### 2.4. Material and Method for Experimental Testing of the Orthopedic Hip Joint Revision Prosthesis

In order to determine the forces at which the two orthopaedic prostheses fail, the EDZ20 hydraulic universal testing machine with a maximum load of 20 tf (200 kN) was used. The tension/compression load is between the fixed crossbar, whose position can be adjusted, and the upper crossbar of the mobile frame driven by a hydraulic motor, which allows it to perform a vertical movement. The intensity of the force is indicated on the graduated dial of the control block throughout the test. When the material fails, the needle indicating the magnitude of the force remains locked at the maximum value (Figure 14).

A digital stereo microscope type INSIZE ISM-PM200SB with a magnification power of 10 … 200× was used to analyse the images obtained after the failure of the orthopaedic systems that were tested. The images are captured using the ISM PRO software and saved in .JPEG format, and the values can be imported into EXCEL or CAD files. It allows the measurement of distances, angles and areas.

Prior to the completion of the orthopaedic prostheses, a fixing device was designed and modelled to allow experimental testing on the EDZ20 universal machine. CAD software, SolidWorks, was used for this, and the components are shown in Figure 15.

These components, the prosthesis elements, the bone component, and the test machine components (in blue) were assembled in SolidWorks and analysed, as shown in Figure 16.

Finally, these components were made from an aluminium alloy. Standard screws and nuts were used to assemble them.

Several methods, techniques, and principles were used to carry out these experiments:-Methods and techniques of experimental research;-Methods and principles of materials strength;-Methods of rigid solids theory and theory of material failure.

### 2.5. Material and Method for Virtual Testing of the Intact and Prosthetic Hip Joint under Normal Human Gait Loading

In this chapter of the paper, we intended to produce a three-dimensional model of the intact and prosthetic hip joint with revision elements. This model was created starting from a set of tomographic images, which was processed and adapted to the prosthetic components. Finally, this model was imported into Ansys Workbench, and three situations were analysed:-The normally intact joint of the human hip;-Joint protected with titanium augment;-Joint protected with morcellated graft and reconstructive mesh.

All these models were virtually tested using the load due to normal human gait.

The transformation of the tomographic images into three-dimensional geometry was done in the first phase with InVesalius software. InVesalius is a free-source medical research software used to generate virtual primary reconstructions of various tissues in the human body. This reconstruction is based on two-dimensional images obtained using computed tomography or MRI; the program generates three-dimensional models virtually corresponding to anatomical parts of the human body. Starting from DICOM images, the software allows the generation of STL (stereolithography) files. These files can also be used for rapid prototyping.

The primary geometry obtained with InVesalius was processed with Geomagic software. This is a program mainly used in reverse engineering, which allows editing and modification of “point clouds” obtained by 3D scanning or CT scanning.

In order to obtain virtual solids, the SolidWorks program was used, which took over the 3D geometry obtained after the processing with Geomagic software. SolidWorks is a computer-aided design and computer-aided engineering program that runs mainly on the Microsoft Windows platform. It is a virtual solids modeller and uses a parametric feature-based approach to create models and assemblies. 

Ansys Workbench software was used to determine the behaviour of the three types of joints. It is a program that uses the finite element method and allows the study of the behaviour of different mechanical or biomechanical systems, which can be shown by results maps. These results maps show the displacement, strain, and stress occurring in the three orthopaedic systems: the normal, the titanium augment prosthetic, and the morcellated graft and reconstructive mesh prosthetic systems.

The Geomagic Capture 3D scanner was used to obtain the models of the prosthesis components with a resolution of minimum 0.08 mm and a minimum scanning distance of 300 mm (Figure 17). The scanning was performed using Geomagic for SolidWorks software. This section may be divided by subheadings. It should provide a concise and precise description of the experimental results, their interpretation, as well as the experimental conclusions that can be drawn.

For the generation of prosthetic components, we used the method of direct visual observation and the method of direct measurement based on the measurement with a calliper with an accuracy of one hundredth of a millimetre. 

Direct and reverse engineering methods, CAD techniques, and the finite element method were also used to determine the behaviour of orthopaedic systems.

### 2.6. Material and Method for Virtual Testing of Normal Hip Joint and Orthopedic Hip Joint Revision Prosthesis

Two software were used for this purpose:SolidWorks, which is a computer-aided design program used in engineering, which allows the generation of multibody models, in this case, for modelling the three systems to be studied using the finite element method;Ansys Workbench, which is a finite element analysis program that allows the study of the behaviour of different mechanical or biomechanical systems and that can show displacement, strain, and stress maps that occur in orthopaedical prostheses that have hip revision prostheses.

CAD methods and techniques were used to obtain the geometries composing the three analysed systems.

The finite element method (FEM) analysis was used to determine the behaviour of these systems [25].

The following simplifying assumptions were used in the analyses of the three orthopaedic systems:Knowing that orthopaedic cement, used in these orthopaedic prostheses, provides stability and fixation of the components, it was not geometrically modelled, and its behaviour was replaced with the use of bonded elements that practically provide cohesion of the elements between themselves [26,27,28,29,30,31,32,33,34,35,36,37,38,39,40,41];It is known that, during the human gait, the variation of the force occurring in the hip joint varies between values of 0 N and about 2300 N. We have considered that it is sufficient to use an equivalent force that has a linear variation between the values of 800 N (characteristic of the orthostatic position) and 2300 N (maximum value of the force during normal human gait [26,27,28,29,30,31,39,42]);For revision prostheses using a morcellated bone graft, the study was considered to be performed after complete osseointegration of the graft, when it acquires the appearance and properties of the trabecular bone [26,27,28,39].

Authors should discuss the results and how they can be interpreted from the perspective of previous studies and of the working hypotheses. The findings and their implications should be discussed in the broadest context possible. Future research directions may also be highlighted.

## 3. Results

### 3.1. Experimental Testing of Orthopaedic Revision Prosthesis with Titanium Augment

The orthopaedic revision prosthesis with titanium augment has been immobilized in the fixing device. In addition, the prosthesis stem was clamped in the upper fixture of the fixation system, which has the possibility to rotate so that the force action is vertical. Figure 18 shows the prosthesis of titanium augment on the EDZ20 universal testing machine.

The force was gradually amplified until the orthopaedic titanium augment prosthesis mechanically failed. The EDZ20 universal machine panel read 1750 Kgf, i.e., 17,167 N. After the mechanical failure of the orthopaedic prosthesis, the prosthesis was studied and analysed with the INSIZE ISM-PM200SB stereomicroscope, and a series of images are shown in Figure 19. The microcracks in the material were analysed.

### 3.2. Experimental Testing of Orthopaedic Revision Prosthesis with Morcellated Graft and Reconstructive Mesh

Similarly, the morcellated graft and reconstructive mesh prosthesis was fixed on the EDZ20 universal machine. At the same time, the verticality of the force application on the prosthesis stem was checked.

The force was gradually amplified from the control panel until the orthopaedic revision prosthesis with morcellated graft and reconstructive mesh failed. The force was read on the display panel and was 1790 Kgf, i.e., 17,559 N. After the mechanical failure of the orthopaedic prosthesis, the prosthesis was analysed with the stereomicroscope, looking at the position, thickness, and size of the microcracks (Figure 20).

### 3.3. Models of the Intact and Prosthetic Hip Joint

To obtain the geometry of the human hip joint, six sets of CT scans on different patients were used in a first phase. These sets were analysed, and finally, the set that was considered to be closest to an ideal hip joint was studied.

Through specific commands that involve the choice of a template that allows the identification, on the basis of shades of grey, of the corresponding bone tissues, a primary model was obtained for the area tomographed on the patient. Obviously, this primary model, shown in Figure 21, also contains elements that are not of interest in this study, including additional geometries due to X-ray reflection and refraction.

To process such a model, which is in STL format, similar to 3D scanning, this geometry was loaded into Geomagic software. Figure 22 shows the interface of the Geomagic software after loading the model.

The STL format, which stores the geometry, consists of the so-called “point cloud”. After retrieving the geometry in Geomagic, the program identified the three closest points and created a triangular spatial surface. At that time, the model contained a very large number, i.e., 2,739,525 primary triangular surfaces (Figure 23).

Obviously, this model was subjected to some processing operations and techniques. Moreover, a number of elements were removed with the intention of finally obtaining the model of an iliac bone (the other being obtained by “mirroring” techniques) and the sacrum. For this purpose, the surfaces of the ribs, vertebrae, the two portions of the femur, and other elements that were not of interest for our study were removed. Surface decimation methods were also used as well as “finishing” operations, which are described in more detail in other papers [26,27,28,29,30,31,32,33,34].

The model, at that time, had 234,830 triangular elementary surfaces. However, it is known that a model with more than 100,000 surfaces cannot be automatically transformed into a virtual solid. In order to reduce this number, “decimation” methods followed by “finishing” techniques were used so as not to affect the quality of the model. Figure 24 shows the models subjected to these techniques and the final model, which had 53,290 elementary surfaces.

This model was loaded into SolidWorks, where it was automatically transformed into a virtual solid. Figure 25 shows the iliac bone model for two visualization options.

For the sacrum, a primary model from Geomagic was used where the other elements were removed, and which comes from the same set of CT scans. This model was subjected to similar operations, and Figure 26 shows the final model of the sacrum bone in SolidWorks.

Because the CT set used to model the iliac bone and sacrum contained only femur fragments, another set was used on a femur taken from a cadaver. Figure 27 shows the InVesalius interface for this second set of CT scans.

This primary geometry in STL format was loaded into Geomagic software for processing and adaptation. At that time, the model contained 1,092,725 triangular primary surfaces. Figure 28 shows the interface of this program with the femur primary geometry loaded.

First of all, this model contained many elements that are not mainly of interest, such as the surfaces of the CT support. These were subjected to some removal operations, and the main steps are shown in Figure 29.

This model has been subjected to operations aiming to decimate the number of elementary surfaces, to “finish” the geometry, and to eliminate self-intersecting surfaces. In the end, the model had 37,919 intermediate surfaces, and it could be transferred into SolidWorks and automatically transformed into a virtual solid. Figure 30 shows this model first in Geomagic, then in SolidWorks.

The virtual bone models were loaded into the Assembly module of SolidWorks. Based on anatomical data, these components were positioned and attached to each other using CAD techniques. Previously, the femur model was scaled and adapted to the other bone components. Additionally, the pubic symphysis was created using different anatomical data. Figure 31 shows the final model of the intact hip joint.

The 3D scanner was used to generate the models of the prosthetic components and basically the same steps were followed as for the bone components. These were then im-ported into the SolidWorks Assembly module. Previously, the bone components were adapted, taking into account the specific surgical techniques of revision prosthetic surgery. Finally, virtual models of the hip joint prosthetic with titanium augment and of the hip joint prosthetic with morcellated graft and re-constructive mesh were obtained. Figure 32 shows these models.

The three models were imported into Ansys Workbench, where we studied their behaviour under normal human gait loading [35,36,37].

### 3.4. Virtual Testing of the Normal (Intact) Hip Joint Subjected to Normal Gait Loading 

The normal (intact) joint model was loaded into Ansys Workbench, and the interface of this software is shown in Figure 33.

First, the division into finite elements was carried out, and 87,854 elements were obtained, having a maximum size of 5 mm and a tetrahedron shape. Figure 34 shows the finite element structure of the analysed system.

In order to perform a static structural analysis, the system was considered to have the fixation surfaces in the femoral condyles area. As previously stated, the force was considered to have a variation between 800 and 2300 N over one second and acting on the sacrum bone, as shown in Figure 35 [29,43,44].

The Ansys program has a module that stores and manages the materials used and their characteristics called engineering data. In this module, a new library was defined, which contains the data contained in Table 1. To determine these values, a selective bibliography was analysed [45,46,47,48,49,50,51,52,53,54,55].

After running the application, maps of the results were obtained. Figure 36 shows the displacement map. Figure 37 shows the strain map.

Figure 38 shows the stress maps for intact hip.

### 3.5. Virtual Testing of the Hip Joint with Titanium Augment Revision Prosthesis Subjected to Normal Gait Loading

The hip joint model with orthopaedic revision prosthesis with titanium augment was loaded into Ansys, as shown in Figure 39.

The division into finite elements and 1,117,700 elements with a maximum size of 5 mm was obtained in the first phase, and we obtained a tetrahedron shape. Figure 40 shows the finite element structure of the orthopaedic system.

Using a static structural analysis, a fixation and loading system similar to the previous situation studied was considered, i.e., fixation in the condylar area of the femoral bones and a force variation between 800 and 2300 N, placed on the sacrum bone.

The materials shown in Table 2 [45,46,47,48,49,50,51,52,53,54,55] were defined and activated in the engineering data module.

After running the simulation, we obtained results maps. Figure 41 shows the displacement map, Figure 42 the strain map, and Figure 43 the stress maps.

### 3.6. Virtual Hip Joint Testing with Revision Prosthesis with Morcellated Graft and Reconstructive Mesh Subjected to Normal Gait Loading 

The hip joint model with revision prosthesis with morcellated bone graft and reconstructive mesh was loaded into Ansys Workbench, as shown in Figure 44.

In a first step, the system was divided into finite elements, resulting in 343,182 elements with a maximum size of 3 mm and a tetrahedron shape, as shown in Figure 45.

In the static structural module, the same loading scheme was used based on the force ranging from 800 to 2300 N positioned on the sacrum bone and the same fixation scheme on the femoral condyles. The required materials were defined or loaded in the engineering data module. These are listed in Table 3 [45,46,47,48,49,50,51,52,53,54,55,56,57].

After running the simulation, we obtained results maps [31,55,56]. Figure 46 shows the displacement map, Figure 47 the strain map, and Figure 48 the stress maps.

We obtained the diagrams in Figure 49 and Figure 50 by analysing the maximum values of the forces at which the revision models mechanically failed.

We also obtained the comparative diagrams in Figure 51, Figure 52 and Figure 53 after analysing the results obtained from the simulations using FEM, which consisted of displacement, strain, and stress maps.

## 4. Discussions

The following observations were highlighted after having analysed the techniques, methods, simulations, and models:The components of the prostheses and revision systems are parameterized so that they can be adapted to different anthropological dimensions;Different “in vitro” analyses can be obtained by attaching these prosthetic revision systems to the virtual bone components;These models can be used in various real or virtual tests and experiments.

The following general observations were also highlighted:Very complicated biological systems can be modelled and simulated by using CAD and FEM methods;The virtual models proposed by this research study have been experimentally validated;Finite element method analysis, coupled with virtual re-construction from CT or MRI images and reverse engineering methods, pave the way for the innovation of orthopaedic systems customized for each patient.

The experimental method presented in this study has certain limitations. One of these is represented by the fact that not all bone components taken from animals can be similar to human ones. However, in those situations, experimental tests can be replaced with virtual ones.

The different techniques known in the engineering field and presented in this work can lead to the improvement of life or can lead to the design of systems or techniques applicable to athletes, to the development of rehabilitation, or endoprosthesis devices [57,58,59,60].

## 5. Conclusions

By studying and analysing the values of the forces under which the two orthopaedic revisions prostheses mechanically failed, the following conclusions and observations can be drawn:The morcellated graft and reconstructive mesh prosthesis was stronger, failing at a force value of 1790 Kgf or 17,560 N;Studying the values of the forces obtained at the mechanical failure of the two orthopaedic prostheses, it was found that there is a minimum difference of 40 Kgf between the two values.

This allows to conclude that both orthopaedic prostheses provide a very good resistance.

By analysing the maximum values obtained from the three simulations using the finite element method, the following conclusions can be drawn:The highest displacement values were obtained in the case of the normal hip joint and the lowest in the case of the titanium augment prosthesis;The maximum strain was obtained in the situation of the normal intact hip joint and the lowest in the situation of the titanium augment;The highest stress values were observed for the titanium augment prosthesis and the lowest for the intact hip joint;The highest values of displacement and strain indicate that the normal intact hip joint is more elastic than the prosthetic hip;Analysing the most stressed areas and surfaces, it was observed that the maximum stresses are found on the prosthesis components, and the bone in contact is less stressed;If the comparative diagrams of the failure forces and maximum stresses in the two orthopaedic revision prostheses are analysed, it is found that the titanium augment prosthesis is the most stressed and consequently failed mechanically at the lowest force, which validates the two virtual and experimental studies.

The work presents virtual and experimental tests that lead to the conclusion that the orthopaedic method based on morcellated graft and reconstructive mesh prosthesis can be used successfully because it has been demonstrated that it provides significant resistance, and the graft allows subsequent bone restoration. In the future, we consider that other orthopaedic systems can be analysed using the techniques presented in the paper. Additionally, using 3D reconstruction techniques, we intend to develop a hardware and software system for diagnosis in virtual reality of various pathological situations in orthopaedics.

## Figures and Tables

**Figure 1 diagnostics-12-01952-f001:**
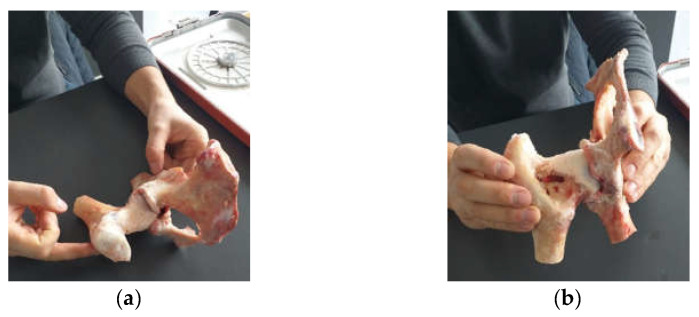
Portion of femur (**a**) and pelvic bone (**b**) taken from an animal (cow).

**Figure 2 diagnostics-12-01952-f002:**
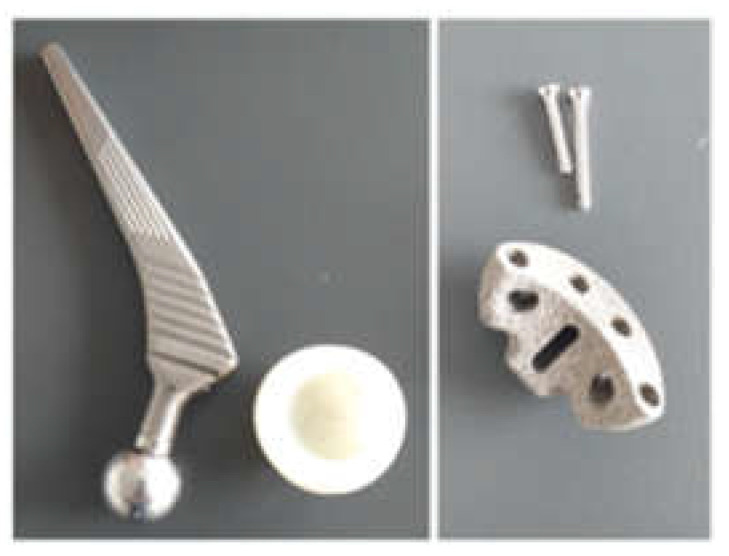
Prosthetic components used for revision prosthesis (prosthesis stem, prosthesis cup, titanium augment, and orthopaedic screws).

**Figure 3 diagnostics-12-01952-f003:**
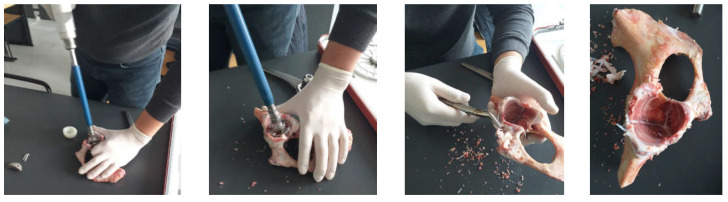
Adaptation of the acetabular cavity to the polyethylene cup (milling, cutting, and removal stages).

**Figure 4 diagnostics-12-01952-f004:**
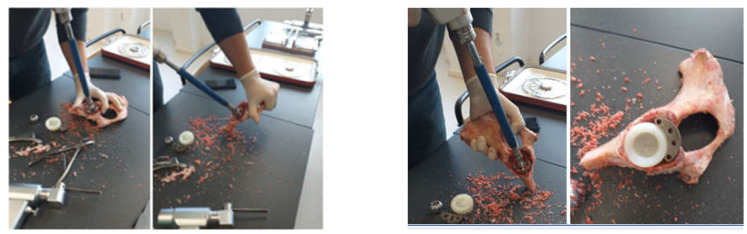
Steps to make the cavity for the titanium augment.

**Figure 5 diagnostics-12-01952-f005:**
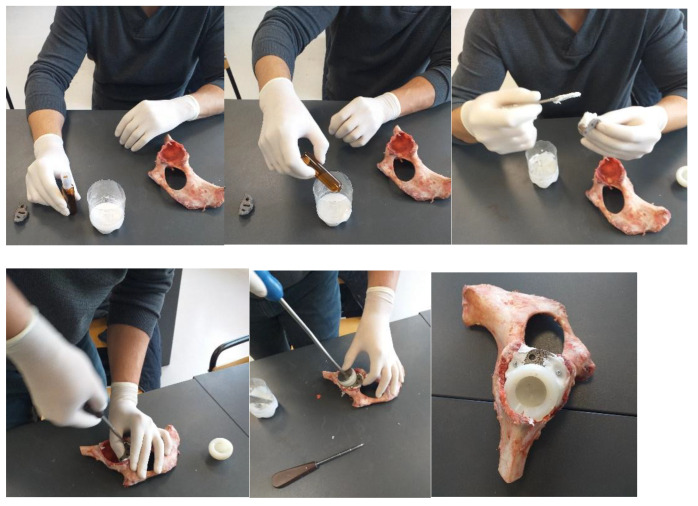
Steps for fixation of prosthesis elements using orthopaedic cement.

**Figure 6 diagnostics-12-01952-f006:**
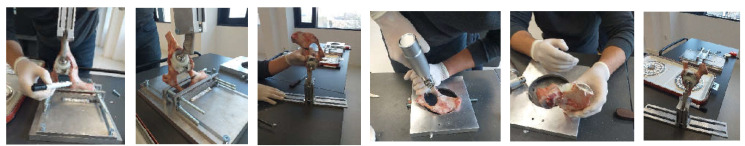
Orthopaedic fitting stages.

**Figure 7 diagnostics-12-01952-f007:**
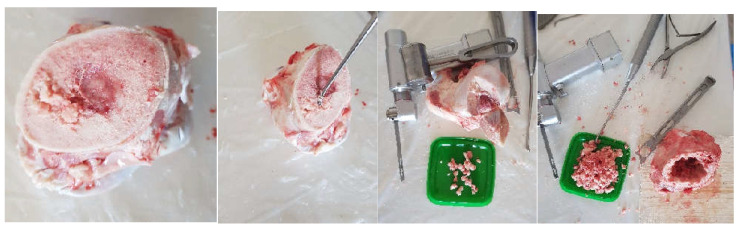
Obtaining the morcellated bone graft.

**Figure 8 diagnostics-12-01952-f008:**
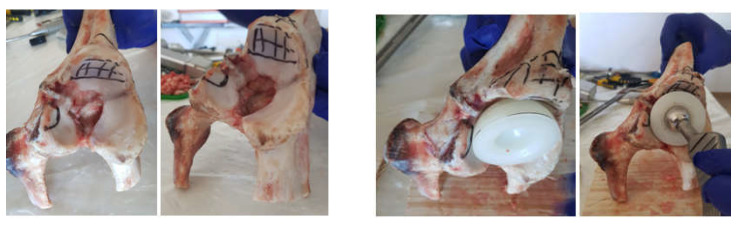
Marking the areas to be removed from the iliac bone.

**Figure 9 diagnostics-12-01952-f009:**
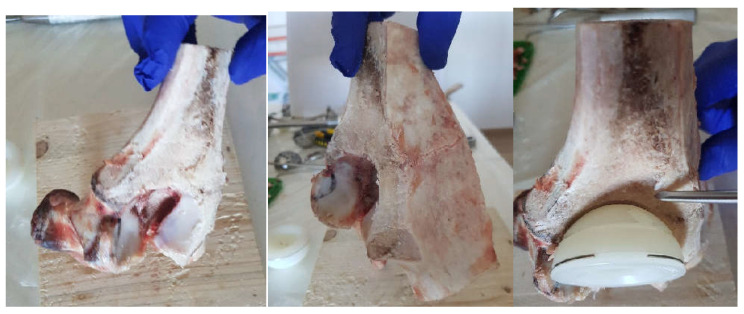
Removing marked areas and positioning the polyethylene cup.

**Figure 10 diagnostics-12-01952-f010:**
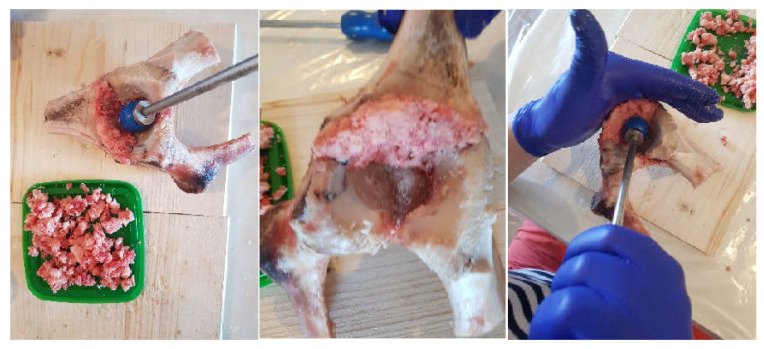
Placement of the bone graft.

**Figure 11 diagnostics-12-01952-f011:**
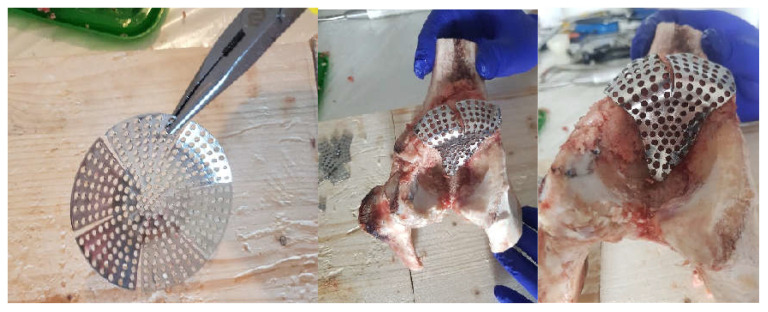
Positioning the reconstructive mesh.

**Figure 12 diagnostics-12-01952-f012:**
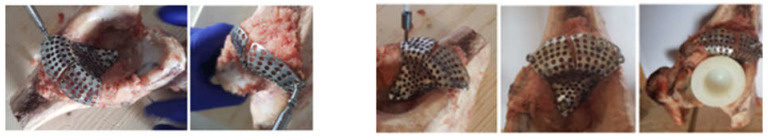
Fixing the reconstructive mesh with orthopaedic screws.

**Figure 13 diagnostics-12-01952-f013:**
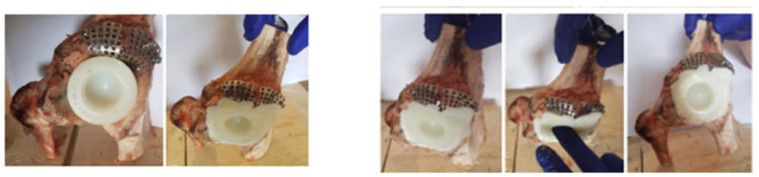
Final model of the orthopaedic prosthesis with morcellated bone graft and reconstructive mesh.

**Figure 14 diagnostics-12-01952-f014:**
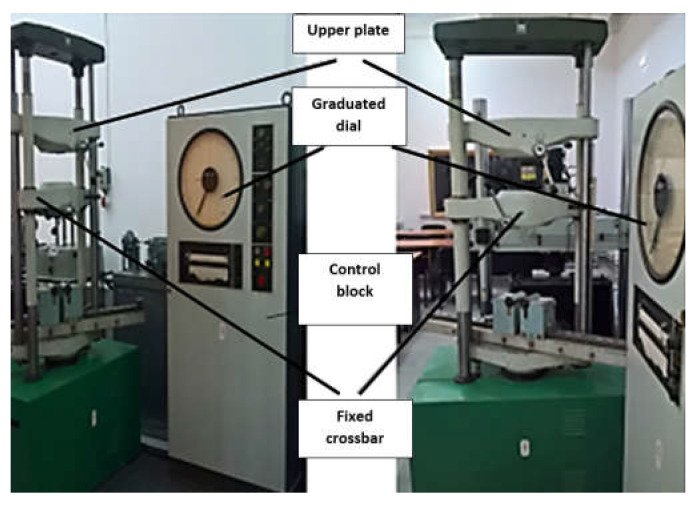
Universal testing machine EDZ20.

**Figure 15 diagnostics-12-01952-f015:**
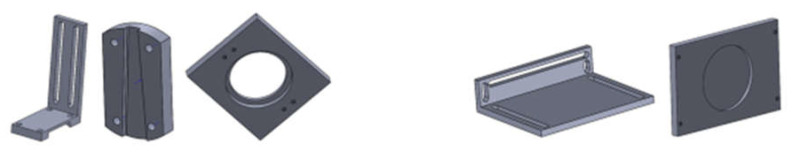
Virtual components of the fixing device.

**Figure 16 diagnostics-12-01952-f016:**
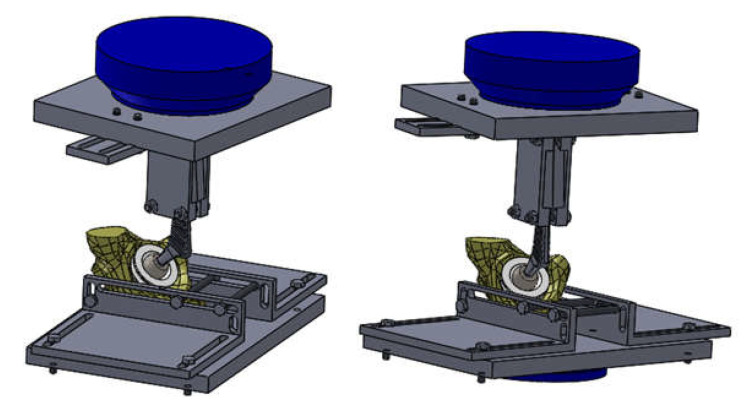
The fixing device (two spatial views).

**Figure 17 diagnostics-12-01952-f017:**
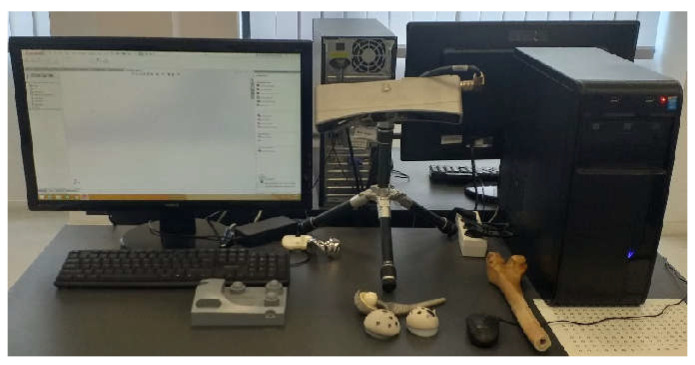
3D Geomagic Capture Scanner.

**Figure 18 diagnostics-12-01952-f018:**
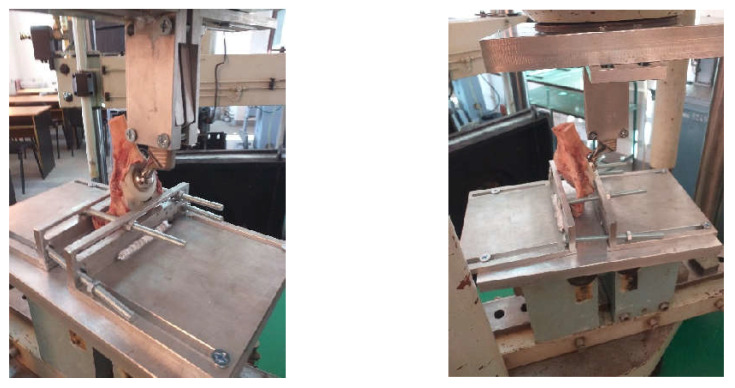
The prosthesis with titanium augment fixed on the EDZ20 machine.

**Figure 19 diagnostics-12-01952-f019:**
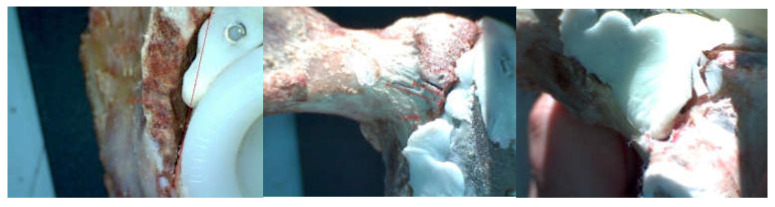
Stereomicroscope images and measurements.

**Figure 20 diagnostics-12-01952-f020:**
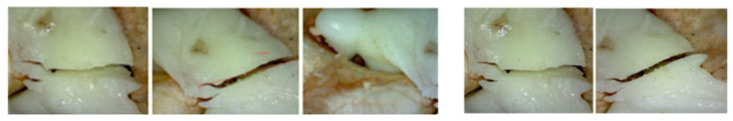
Micro fissures in the orthopaedic prosthesis analysed under the microscope.

**Figure 21 diagnostics-12-01952-f021:**
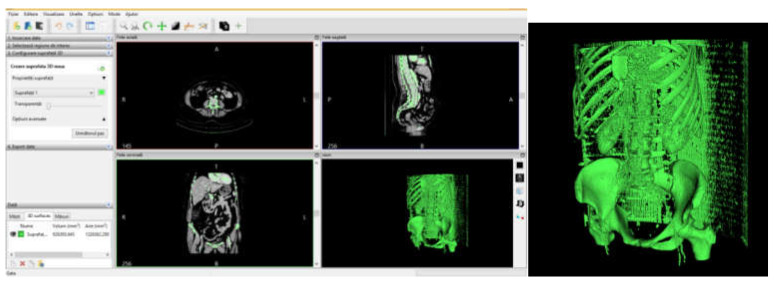
Primary geometry obtained in the InVesalius software.

**Figure 22 diagnostics-12-01952-f022:**
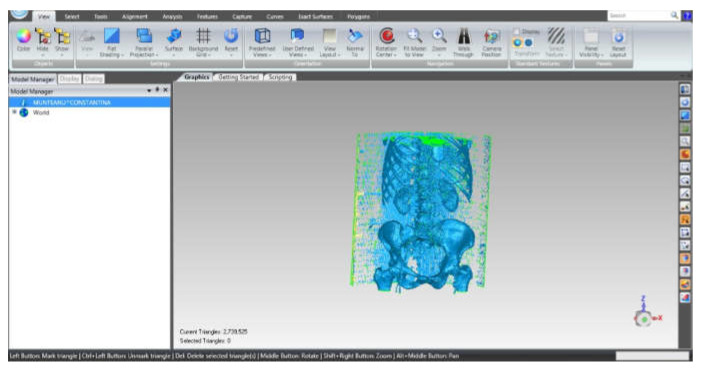
Geomagic software interface.

**Figure 23 diagnostics-12-01952-f023:**
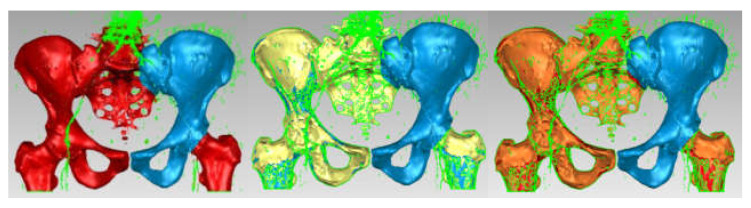
Steps in Geomagic for obtaining the left iliac bone.

**Figure 24 diagnostics-12-01952-f024:**
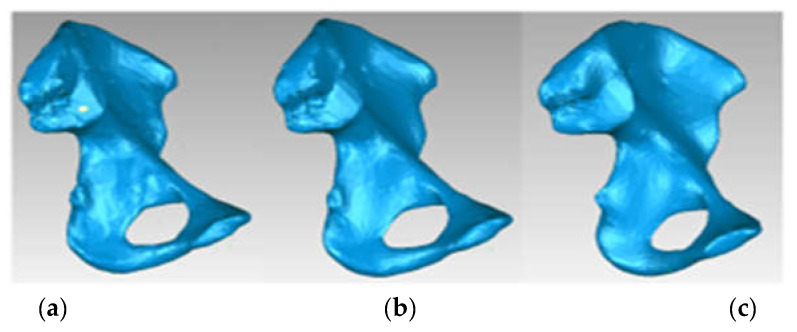
“Decimation” and “finishing” stages applied in Geomagic (**a**) “decimation”; (**b**) first “finishing” stage; and (**c**) last “finishing” stage—final model.

**Figure 25 diagnostics-12-01952-f025:**
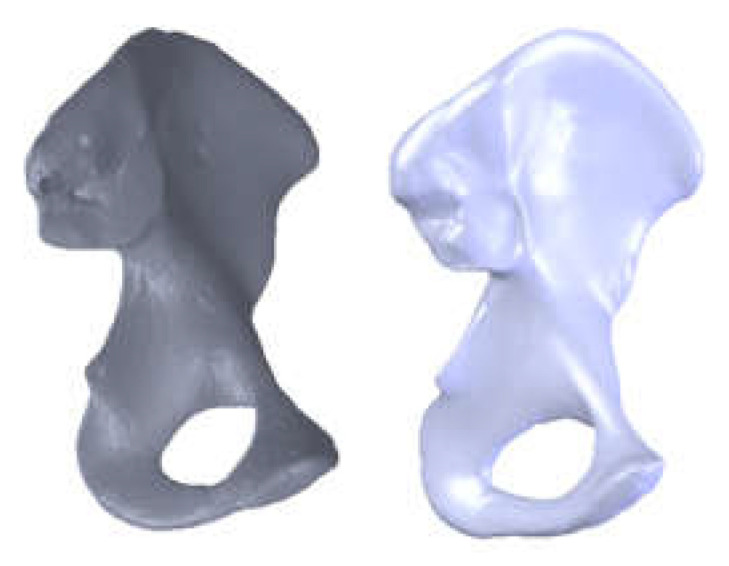
Left iliac bone model (two views with different viewing options).

**Figure 26 diagnostics-12-01952-f026:**
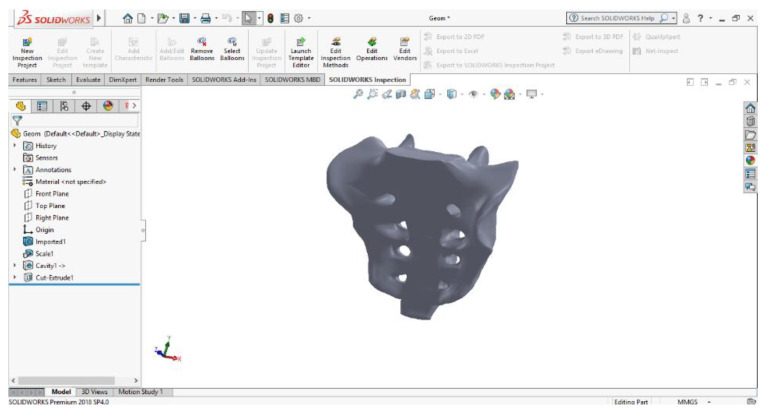
Final model of the sacrum in SolidWorks.

**Figure 27 diagnostics-12-01952-f027:**
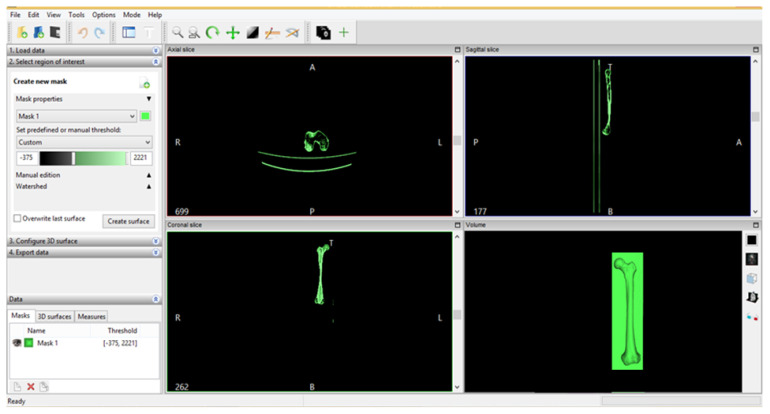
InVesalius interface for the second set of CT scans.

**Figure 28 diagnostics-12-01952-f028:**
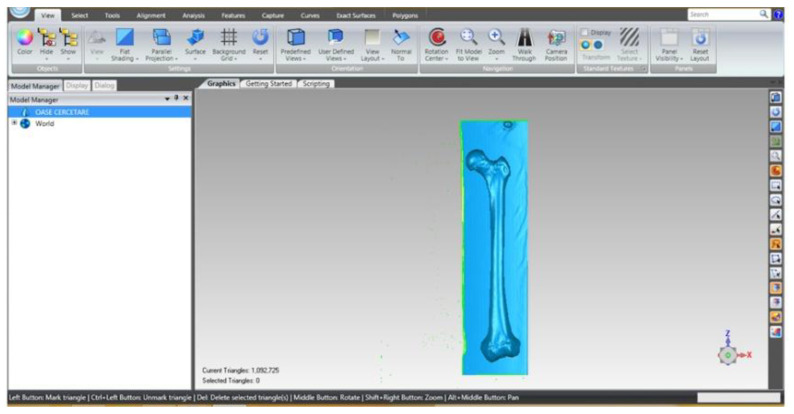
Primary geometry of the femur in Geomagic.

**Figure 29 diagnostics-12-01952-f029:**
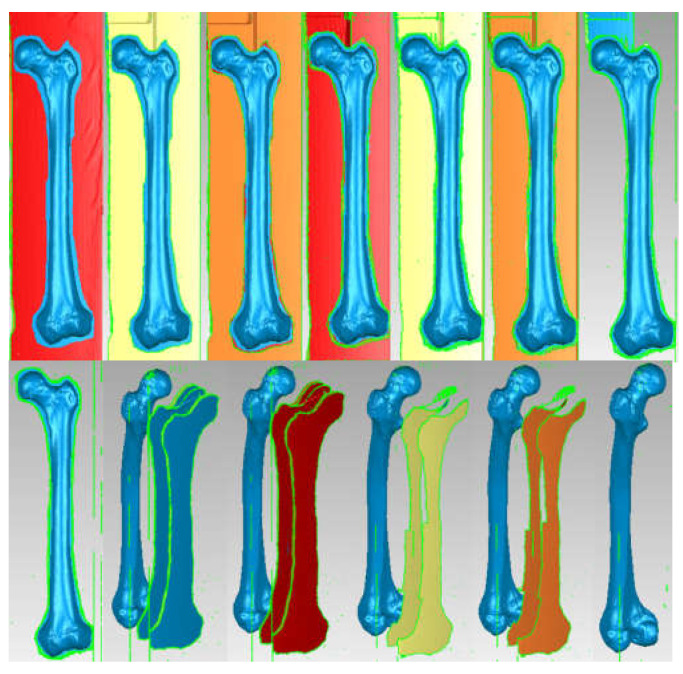
Steps to remove additional surfaces in Geomagic.

**Figure 30 diagnostics-12-01952-f030:**
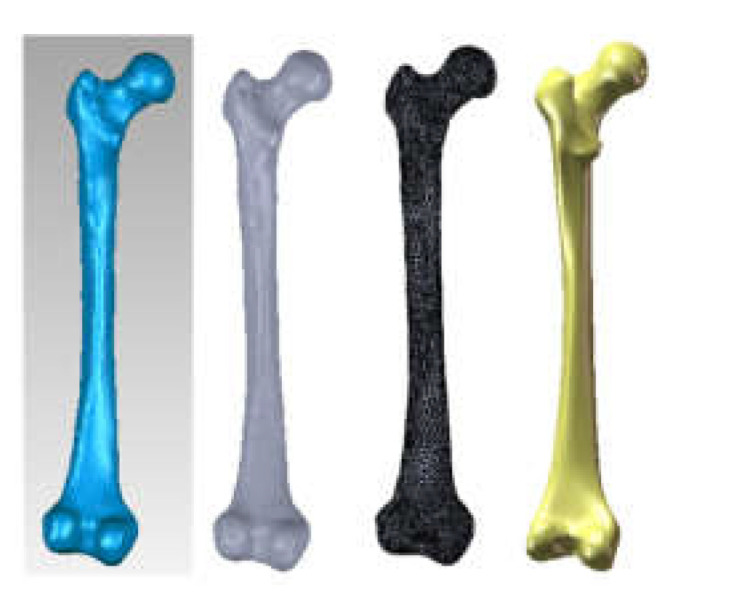
The model of the femur first in Geomagic, then in SolidWorks.

**Figure 31 diagnostics-12-01952-f031:**
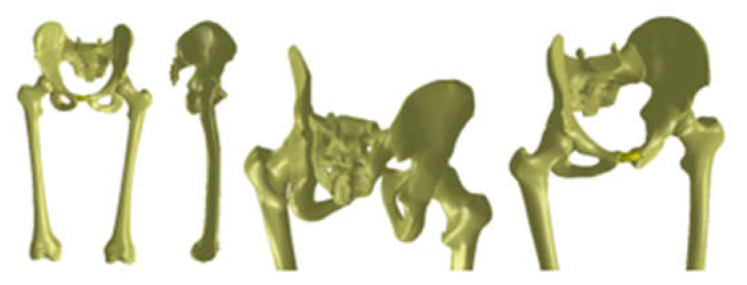
Final geometry of the intact hip joint in SolidWorks.

**Figure 32 diagnostics-12-01952-f032:**
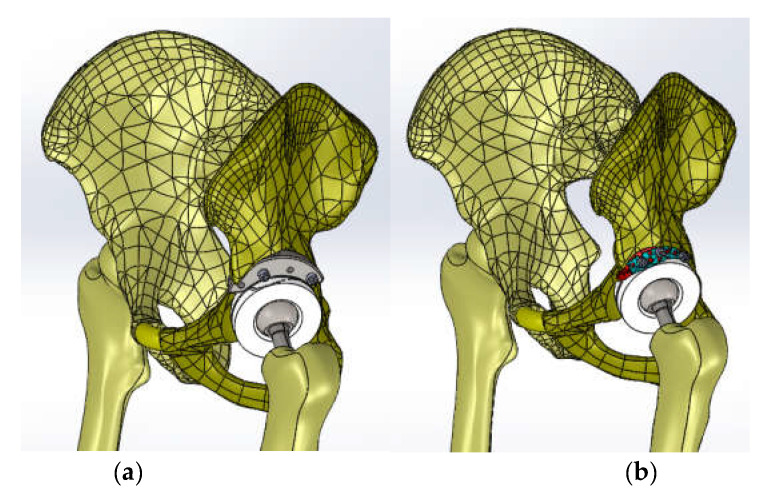
Final geometry of the hip joint with revision prosthesis in SolidWorks. (**a**) Orthopaedic prosthesis with titanium augment. (**b**) Orthopaedic prosthesis with morcellated graft and reconstructive mesh.

**Figure 33 diagnostics-12-01952-f033:**
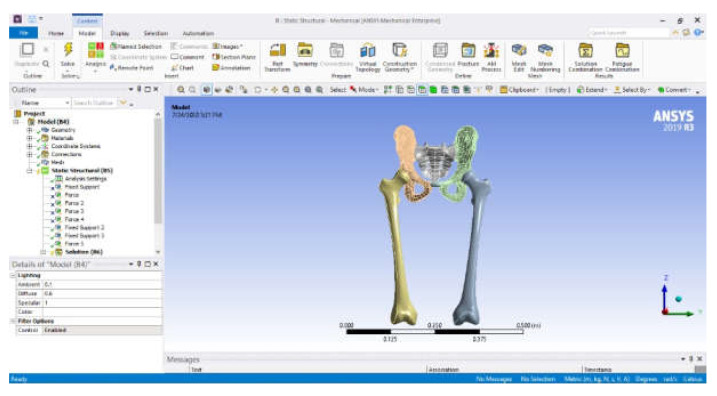
Ansys Workbench software interface after loading the normal human hip joint model.

**Figure 34 diagnostics-12-01952-f034:**
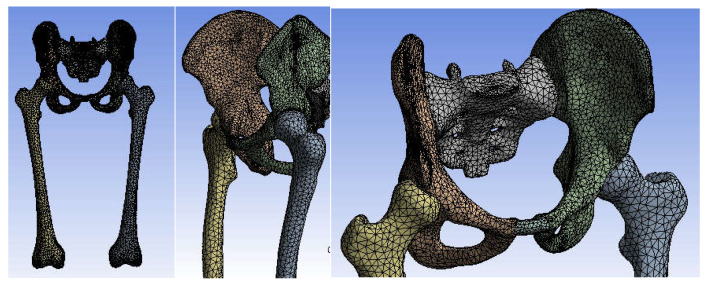
Finite element structure of the system under analysis.

**Figure 35 diagnostics-12-01952-f035:**
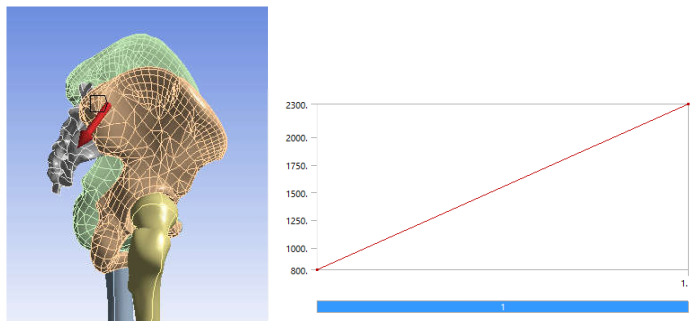
Position and variation of the force acting on the studied system.

**Figure 36 diagnostics-12-01952-f036:**
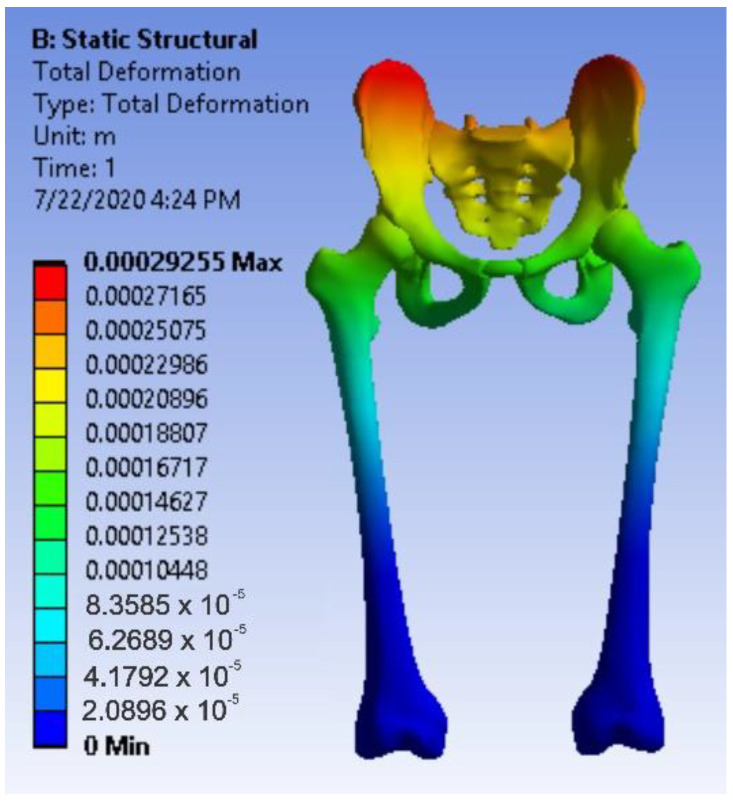
Displacement map for intact hip.

**Figure 37 diagnostics-12-01952-f037:**
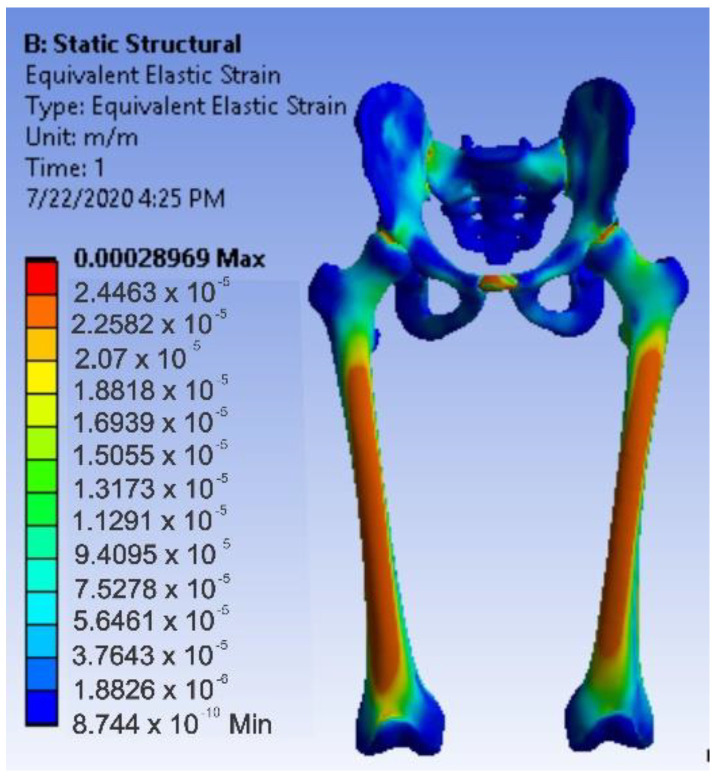
Strain map for intact hip.

**Figure 38 diagnostics-12-01952-f038:**
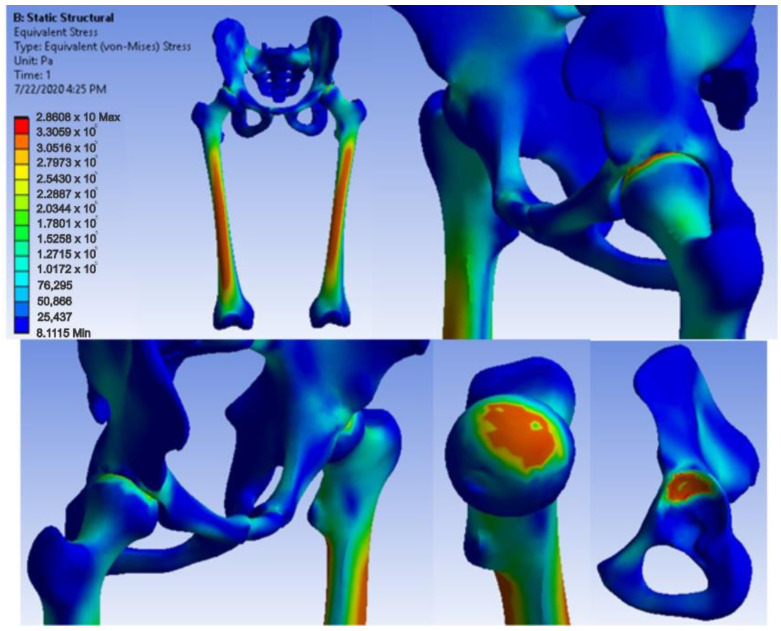
Stress maps for intact hip.

**Figure 39 diagnostics-12-01952-f039:**
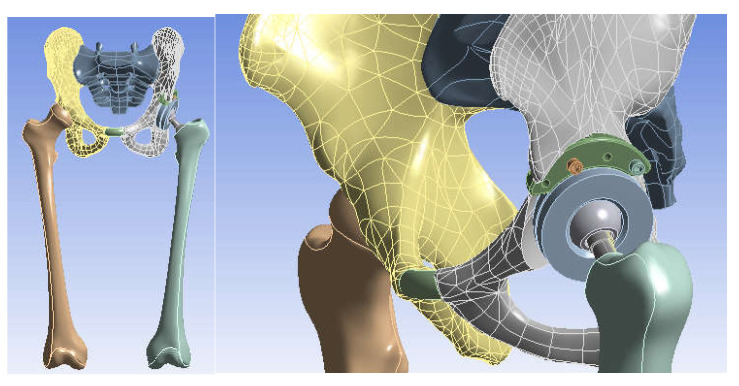
Analysed orthopaedic model loaded in Ansys.

**Figure 40 diagnostics-12-01952-f040:**
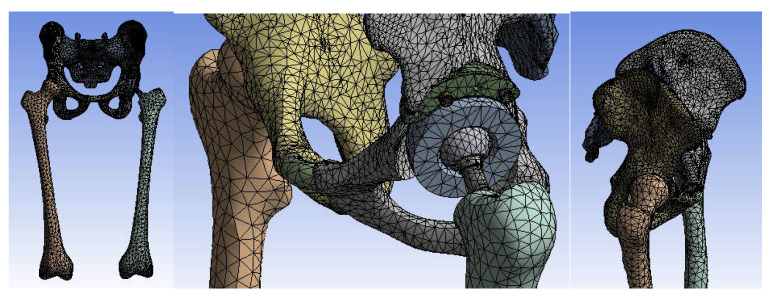
Finite element structure of the analysed orthopaedical system.

**Figure 41 diagnostics-12-01952-f041:**
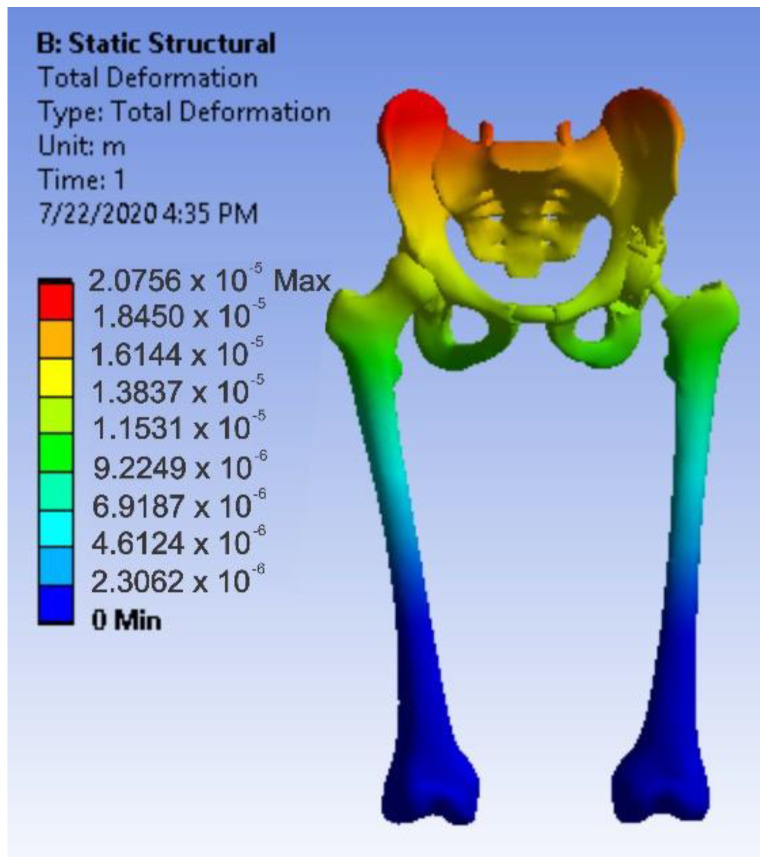
Displacement map for revision prosthesis with titanium augment.

**Figure 42 diagnostics-12-01952-f042:**
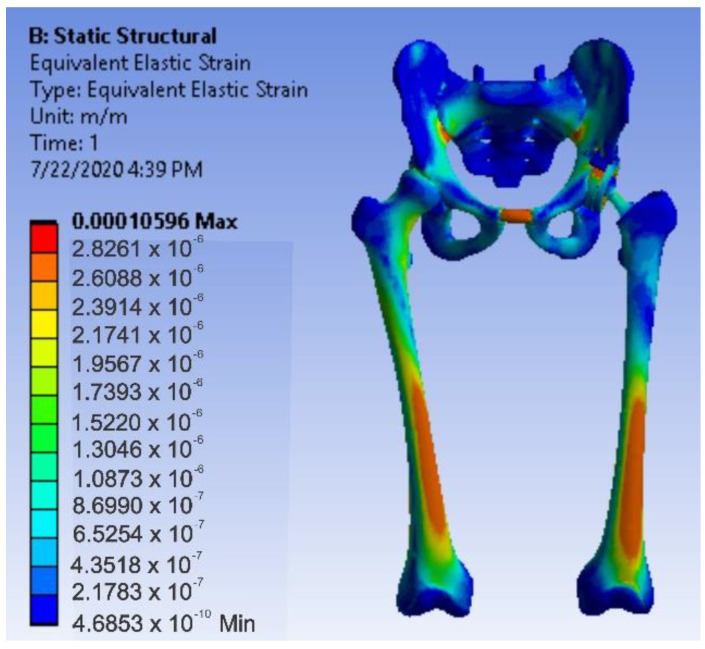
Strain map for revision prosthesis with titanium augment.

**Figure 43 diagnostics-12-01952-f043:**
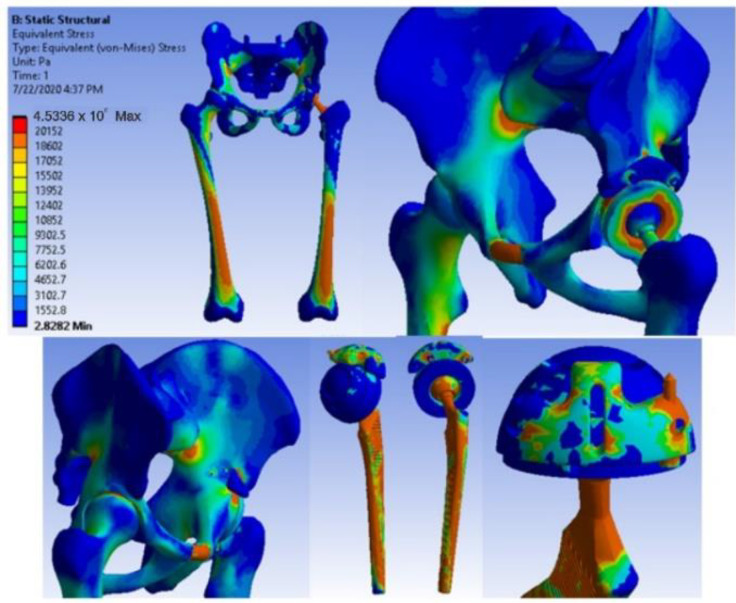
Stress maps for revision prosthesis with titanium augment.

**Figure 44 diagnostics-12-01952-f044:**
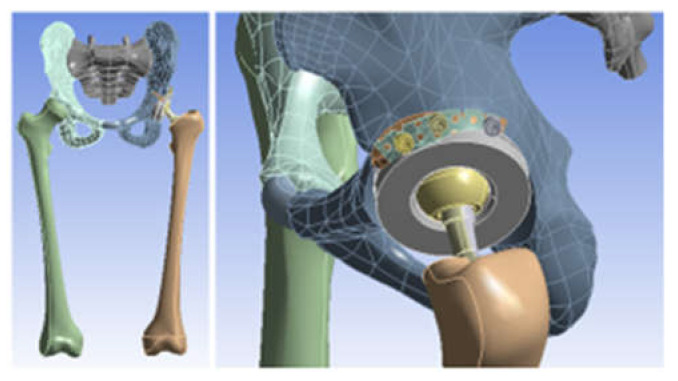
The analysed model loaded in Ansys Workbench.

**Figure 45 diagnostics-12-01952-f045:**
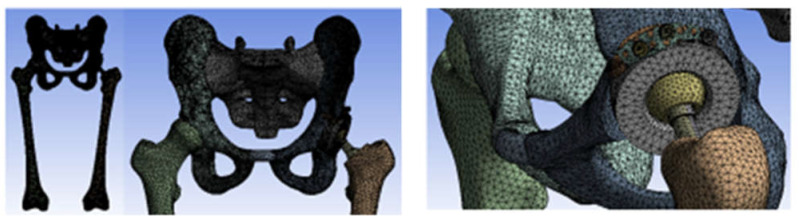
The finite element structure of the analysed orthopaedic system.

**Figure 46 diagnostics-12-01952-f046:**
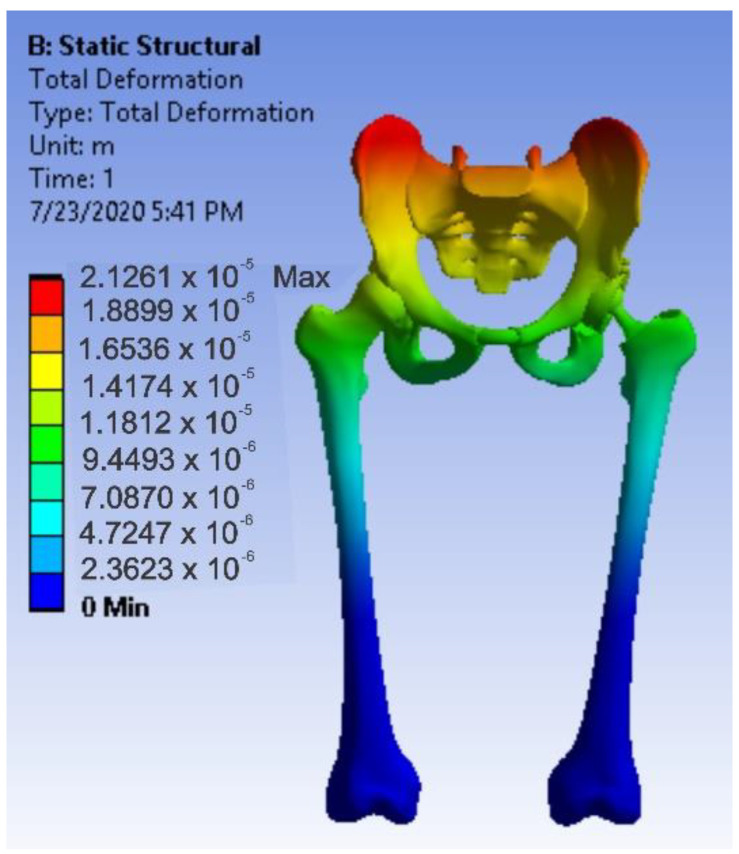
Displacement map for the system with morcellated bone graft.

**Figure 47 diagnostics-12-01952-f047:**
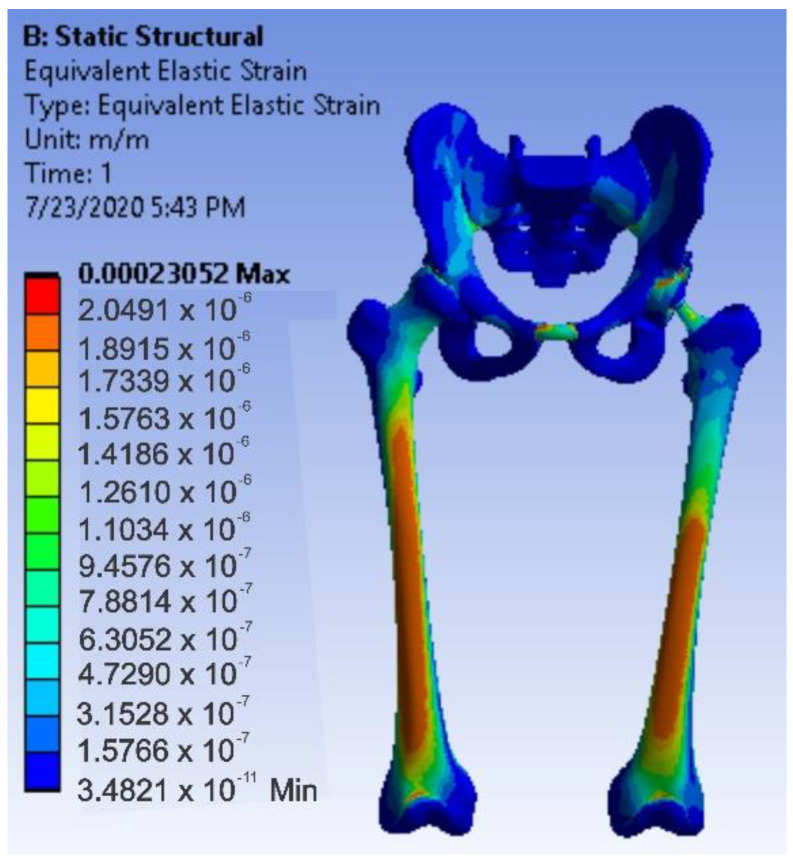
Strain map for the system with morcellated bone graft.

**Figure 48 diagnostics-12-01952-f048:**
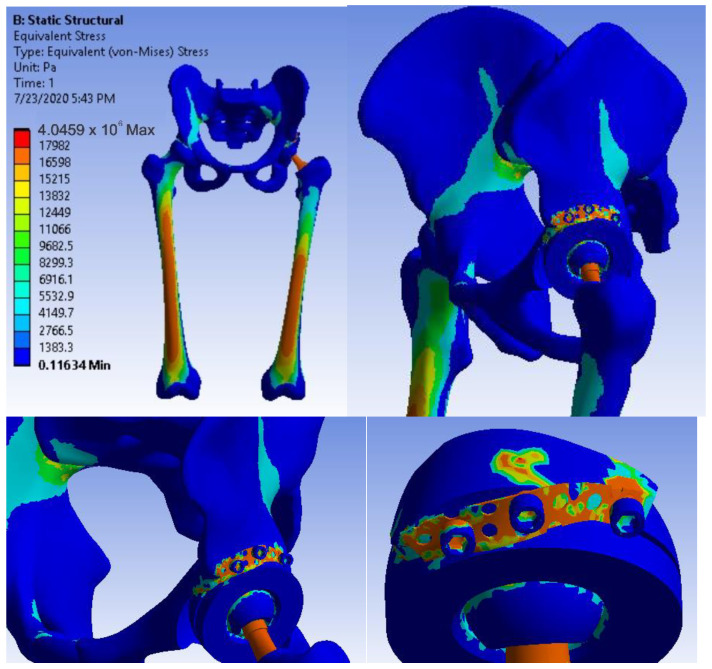
Stress maps for the system with morcellated bone graft.

**Figure 49 diagnostics-12-01952-f049:**
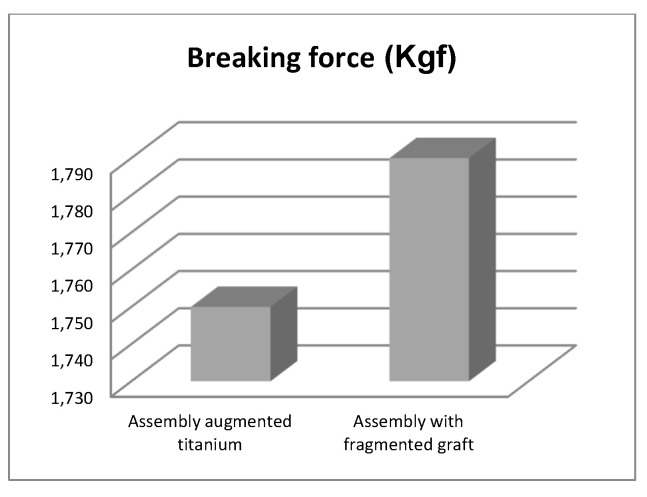
Comparative diagram of breaking forces in Kgf.

**Figure 50 diagnostics-12-01952-f050:**
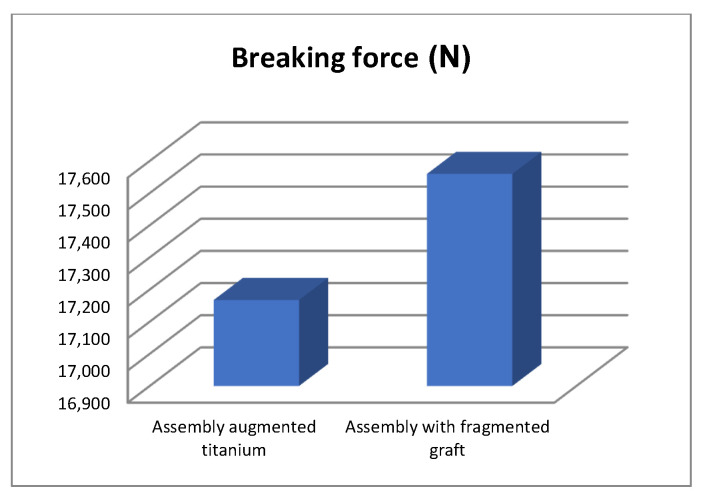
Comparative diagram of breaking forces in N.

**Figure 51 diagnostics-12-01952-f051:**
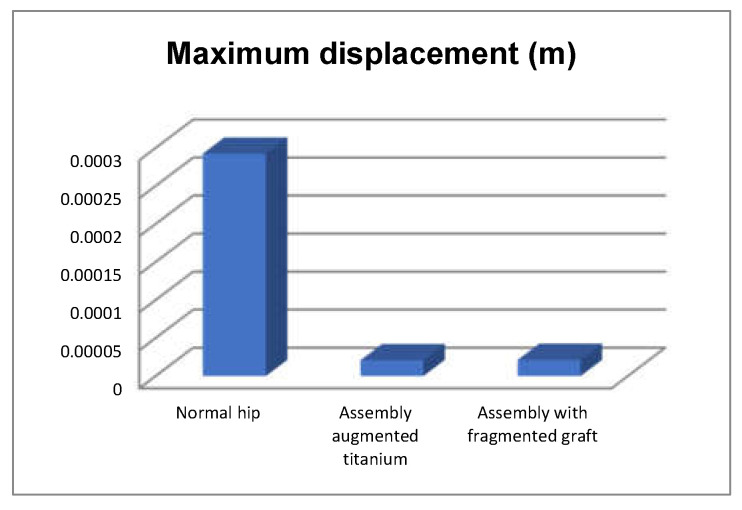
Comparative diagram of maximum displacements.

**Figure 52 diagnostics-12-01952-f052:**
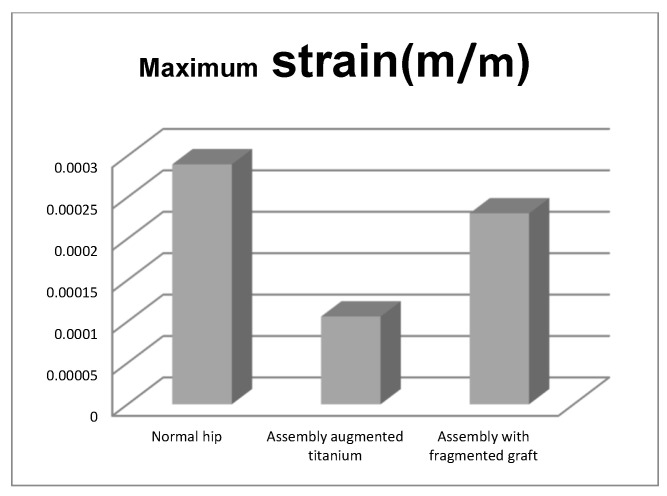
Comparative diagram of maximum strains.

**Figure 53 diagnostics-12-01952-f053:**
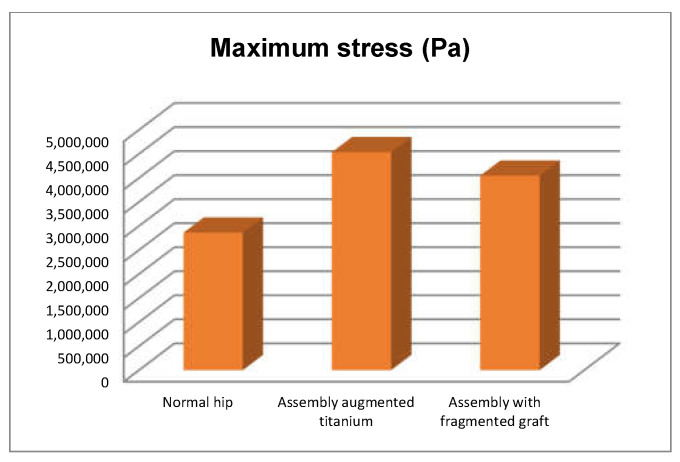
Comparative diagram of maximum stress.

**Table 1 diagnostics-12-01952-t001:** Physico-mechanical characteristics of the materials used in the simulation.

Component	Material	Density (Kg/m^3^)	Young’s Modulus (GPa)	Poisson Ratio
Femur, pelvis, sacrum	Bone	1400	1 × 10^10^	0.3
Pubic symphysis	Ligament	955	1.2 × 10^9^	0.42

**Table 2 diagnostics-12-01952-t002:** Physico-mechanical characteristics of the materials used in the simulation for the studied orthopaedic system.

Component	Material	Density (Kg/m^3^)	Young’s Modulus (GPa)	Poisson Ratio
Femur, pelvis, sacrum	Bone	1400	1 × 10^10^	0.3
Pubic symphysis	Ligament	955	1.2 × 10^9^	0.42
Augment	Titanium alloy	4620	9.6 × 10^10^	0.36
Polyethylene cup	Polyethylene	950	1.1 × 10^9^	0.42
Spherical metal head	Stainless steel	7750	1.93 × 10^11^	0.31
Orthopaedic screw	Stainless steel	7750	1.93 × 10^11^	0.31
Femoral stem	Stainless steel	7750	1.93 × 10^11^	0.31

**Table 3 diagnostics-12-01952-t003:** Physico-mechanical characteristics of the materials used in the simulation for the system with morcellated bone graft.

Component	Material	Density (Kg/m^3^)	Young’s Modulus (GPa)	Poisson Ratio
Femur, pelvis, sacrum	Bone	1400	1 × 10^10^	0.3
Morcellated bone graft	Trabecular bone	2140	1.76 × 10^10^	0.25
Pubic symphysis	Ligament	955	1.2 × 10^9^	0.42
Reconstruction mesh	Stainless steel	7750	1.93 × 10^11^	0.31
Polyethylene cup	Polyethylene	950	1.1 × 10^9^	0.42
Spherical metal head	Stainless steel	7750	1.93 × 10^11^	0.31
Orthopaedic screw	Stainless steel	7750	1.93 × 10^11^	0.31
Femoral stem	Stainless steel	7750	1.93 × 10^11^	0.31

## Data Availability

The authors declare that the data of this research are available from the corresponding authors upon reasonable request.

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
