# Peer review of "An Experimental and Virtual Approach to Hip Revision Prostheses"

_diagnostics, 2022, doi:10.3390/diagnostics12081952_

Round 1

Reviewer 1 Report

Dear Authors, 

you made a great work! 

However, some improvements are mandatory before acceptance. 

Author Response

Dear Reviewer,

I start by thanking you for your appreciations and observations.

I made the following changes to the manuscript:

  1. I modified the abstract to be clearer and used the indications from the MDPI templates. I also added keywords.
  2. I modified “of joint com-ponents and bone loss” to “of joint components and bone loss”
  3. I modified “of fac-tors” to “factors”
  4. The "Material and Method" chapter was divided into several parts.
  5. I improve the quality of old figures 1, 7 and 19.
  6. The chapters of the work have been modified. Now, the manuscript has the following chapters: Introduction, Material and Method, Results, Discussion and Conclusions.

I end here, with the hope that the answers and changes made to the manuscript are satisfactory for you.

Thank you once again for your cooperation!

8/2/2022

Assoc.prof. Dragos-Laurentiu Popa

Reviewer 2 Report

This manuscript entitled “An Experimental and Virtual Approach to Hip Revision Prostheses” primarily aimed to investigate the hip revision protheses via a method which combined the mechanical test and finite element analysis. The authors bring an interesting study, but there are still some problems that cannot up this review to a publishing level. Some suggestions are listed in the specific comments below.

Specific comments:

1.     In the abstract part, it is recommended to add the aim of this study.

2.     In the abstract part, please add your main findings and highlights of your study.

3.     In the introduction part, line 32-34, “The hip arthroplasty is a therapeutic solution and can be caused, most frequently, by primary and secondary coxarthrosis, due to or followed by traumatic conditions.” Please add more references here.

4.     In the introduction part, line 47-53, “The hip revision surgery involves…reconstruct or restore additional metal elements” Please add more references here.

5.     In the introduction part, last paragraph. In the opinion of reviewer, you should mention the research gap and highlights of your research instead of the method you used.

6.     The introduction part of your study is not well structured. You do not need to separate it into many paragraphs. Please improve it.

7.     In the Performing Orthopedic Hip Revisions part, material and method, “Their mechanical properties were also considered to be similar to human ones”, please cite references here to support this statement.

8.     In the Virtual testing of normal hip joint and orthopaedic hip joint revision prosthesis part, material and method, line 391-397, In the opinion of reviewer, you do not need to mention the history of finite element analysis.

9.     In the discussion and conclusions part, in the opinion of reviewer, the discussion is not detailed enough. Besides, figures should not be put in the discussion part. I think it belongs to the results of your study. Some recently studies could be added in the discussion, such as:

Gait Synergy Analysis and Modeling on Amputees and Stroke Patients for Lower Limb Assistive Devices. Sensors 2022, 22, 4814. https://doi.org/10.3390/s22134814

Relationship Between Isometric Hip Torque With Three Kinematic Tests in Soccer Players. Physical Activity and Health, 4(1), 142–149. DOI: http://doi.org/10.5334/paah.65

Comparison of femur strain under different loading scenarios: Experimental testing. Proceedings of the Institution of Mechanical Engineers, Part H: Journal of Engineering in Medicine. 2021;235(1):17-27. doi:10.1177/0954411920951033

10.  It is recommended to add a separate paragraph to describe the conclusion which shows detailed findings about this manuscript as well as what are the contributions for future clinical or scientific research.

11.  What are the limitations of this study? Please provide relevant description.

12.  There are so many figures in this manuscript. Many of those are unnecessary, such as figure 1, 2, 3, 18-22, and many figures about interface of software, please delete them. Besides, the pixel in all figures is too blurred, please replace clearer figures.

13.  Please do check the language and grammar mistakes throughout the whole article to further improve clarity.

Author Response

Dear Reviewer,

Thank you very much for your cooperation and your comments.

I made the following changes to the manuscript:

  1. In the abstract part, I add the aim of this study.
  2. In the abstract part, I add main findings and highlights of this study.
  3. In the introduction part, for the phrase “The hip arthroplasty is a therapeutic solution and can be caused, most frequently, by primary and secondary coxarthrosis, due to or followed by traumatic conditions.” I add references [2,3]
  4. In the introduction part, for the phrase “The hip revision surgery involves…reconstruct or restore additional metal elements” I add references [6-8].
  5. In the introduction part, in the last paragraph, I add highlights of our research.
  6. In the introduction part I reduced the number of paragraphs.

7    For the phrase, “Their mechanical properties were also considered to be similar to human ones”, I add references [19-20].

  1. In the Virtual testing of normal hip joint and orthopaedic hip joint revision prosthesis part, material and method, I remove the history of finite element analysis.
  2. First, the discussion and conclusions part I divided in two chapters, first ”Discussion” and second, “Conclusions” and I moved the figures in “Results” chapter. Also, I made some remarks about the three studies you mentioned and I add references [51-53].
  3. In the end of “Conclusions” chapter I add future work of our team, related to this paper.
  4. In the end of “Discussion” chapter I add the limitations of this study.
  5. I removed figures 2,18, 21 and 24 and I improve the quality of some figures.
  6. As much as possible, I have checked for language and grammar mistakes to further improve clarity.

I end here, with the hope that the answers and changes made to the manuscript are satisfactory for you. Thank you once again for your cooperation!

8/2/2022

Assoc.prof. Dragos-Laurentiu Popa

Round 2

Reviewer 2 Report

The authors have made a good revision, I recommend to accept now. 

Author Response

Dear Reviewer,

I start by thanking you for your appreciations and observations.

I have included in the References chapter four references suggested by your Journal, such as:

  1. K. Ciliberti, G. Cesarelli, L. Guerrini, A.E. Gunnarsson, R. Forni, R. Aubonnet, M. Recenti, D. Jacob, H. Jr. Jónsson, V. Cangiano, A.S. Islind, M. Gambacorta and P. Gargiulo, The role of bone mineral density and cartilage volume to predict knee cartilage degeneration, Eur J Transl Myol., 32(2) (2022) 1-9.
  2. Recenti, C. Ricciardi, K. Edmunds, D. Jacob, M. Gambacorta and P. Gargiulo, Testing soft tissue radiodensity parameters interplay with age and self-reported physical activity, Eur J Transl Myol., 31(3) (2021) 1-7.
  3. Latessa, C. Ricciardi, D. Jacob, H. Jr. Jónsson, M. Gambacorta, G. Improta and P. Gargiulo, Health technology assessment through Six Sigma Methodology to assess cemented and uncemented protheses in total hip arthroplasty, Eur J Transl Myol., 31(1) (2021) 1-11.
  4. Ricciardi, H Jr. Jónsson, D. Jacob, G. Improta, M. Recenti, M.K. Gíslason, G. Cesarelli, L. Esposito, V. Minutolo, P. Bifulco and P. Gargiulo, Improving Prosthetic Selection and Predicting BMD from Biometric Measurements in Patients Receiving Total Hip Arthroplasty, Diagnostics (Basel), 10(10) (2020) 1-11.

In some sentences, I have reformulated some expressions and terms in English for better understanding and coherence.

I end my letter here, with the hope that the answers and changes made to the manuscript are satisfactory for you.

Thank you once again for your cooperation!

8/8/2022

Assoc.prof. Dragos-Laurentiu Popa